# Machine Learning-Based Automatic Classification of Video Recorded Neonatal Manipulations and Associated Physiological Parameters: A Feasibility Study

**DOI:** 10.3390/children8010001

**Published:** 2020-12-22

**Authors:** Harpreet Singh, Satoshi Kusuda, Ryan M. McAdams, Shubham Gupta, Jayant Kalra, Ravneet Kaur, Ritu Das, Saket Anand, Ashish Kumar Pandey, Su Jin Cho, Satish Saluja, Justin J. Boutilier, Suchi Saria, Jonathan Palma, Avneet Kaur, Gautam Yadav, Yao Sun

**Affiliations:** 1Child Health Imprints (CHIL) Pte. Ltd., Singapore 048545, Singapore; shubham@childhealthimprints.com (S.G.); iamjayantkalra@gmail.com (J.K.); ravneet@childhealthimprints.com (R.K.); ritu@childhealthimprints.com (R.D.); 2Department of Pediatrics, Kyorin University, Tokyo 181-8612, Japan; kusuda-satoshi@umin.ac.jp; 3Department of Pediatrics, University of Wisconsin School of Medicine and Public Health, Madison, WI 53726, USA; mcadams@pediatrics.wisc.edu; 4Department of Computer Science and Engineering, Indraprastha Institute of Information Technology, New Delhi 110020, India; anands@iiitd.ac.in; 5Department of Mathematics, Indraprastha Institute of Information Technology, New Delhi 110020, India; ashish.pandey@iiitd.ac.in; 6College of Medicine, Ewha Womans University Seoul, Seoul 03760, Korea; sujin-cho@ewha.ac.kr; 7Department of Neonatology, Sir Ganga Ram Hospital, New Delhi 110060, India; satishsaluja@gmail.com; 8Department of Industrial and Systems Engineering, College of Engineering, University of Wisconsin, Madison, WI 53706, USA; jboutilier@wisc.edu; 9Machine Learning and Healthcare Lab, Johns Hopkins University, 3400 N. Charles St, Baltimore, MD 21218, USA; ssaria@bayesianhealth.com; 10Department of Pediatrics, Stanford University, Stanford, CA 94305, USA; JPalma@stanfordchildrens.org; 11Department of Neonatology, Apollo Cradle Hospitals, New Delhi 110015, India; avneet.raveen@gmail.com; 12Department of Pediatrics, Kalawati Hospital, Rewari 123401, India; gautam2880@gmail.com; 13Division of Neonatology, University of California, San Francisco, CA 92521, USA; Yao.Sun@ucsf.edu

**Keywords:** CNN, electronic medical records, IoT, LSTM, machine learning, neonatal intensive care units, physiological deviations, physiological parameters, streaming server, video monitoring

## Abstract

Our objective in this study was to determine if machine learning (ML) can automatically recognize neonatal manipulations, along with associated changes in physiological parameters. A retrospective observational study was carried out in two Neonatal Intensive Care Units (NICUs) between December 2019 to April 2020. Both the video and physiological data (heart rate (HR) and oxygen saturation (SpO_2_)) were captured during NICU hospitalization. The proposed classification of neonatal manipulations was achieved by a deep learning system consisting of an Inception-v3 convolutional neural network (CNN), followed by transfer learning layers of Long Short-Term Memory (LSTM). Physiological signals prior to manipulations (baseline) were compared to during and after manipulations. The validation of the system was done using the leave-one-out strategy with input of 8 s of video exhibiting manipulation activity. Ten neonates were video recorded during an average length of stay of 24.5 days. Each neonate had an average of 528 manipulations during their NICU hospitalization, with the average duration of performing these manipulations varying from 28.9 s for patting, 45.5 s for a diaper change, and 108.9 s for tube feeding. The accuracy of the system was 95% for training and 85% for the validation dataset. In neonates <32 weeks’ gestation, diaper changes were associated with significant changes in HR and SpO_2_, and, for neonates ≥32 weeks’ gestation, patting and tube feeding were associated with significant changes in HR. The presented system can classify and document the manipulations with high accuracy. Moreover, the study suggests that manipulations impact physiological parameters.

## 1. Introduction

Worldwide, of the 150 million annual births, 15 million are preterm neonates [1]. Advances in neonatal care have improved clinical outcomes [2,3,4]. Digitization of Neonatal Intensive Care Unit (NICU) workflow using the electronic medical record (EMR) and medical device physiological data has enhanced integration and data utilization in analyzable format. [5]. In theory, automated data entry and monitoring reduce the clinical staff’s manual workload and allow more time to focus on patient care [6]. Improved NICU digital infrastructure has resulted in the generation of multi-modal temporal databases, such as Medical Information Mart for Intensive Care (MIMIC), which encapsulate the integrated “big data” of the ICU environment [7,8]. 

Retrospective studies have demonstrated the relationship of physiological signal variations with the onset of diseases [9,10,11]. A loss of heart rate variability (HRV) in the early hours after birth is associated with high morbidity in newborns [12]; Heart Rate Onservation (HeRO) monitoring has demonstrated subtle irregularities in HRV as an early indicator of disease [13]. Furthermore, studies have compared physiological signals immediately before, during, and after performing procedures/manipulations on neonates [14,15,16]. These manipulations include invasive procedures, such as intubation, peripheral intravenous line insertion, and common non-invasive handling, of neonates for care, such as patting, diaper change, and feeding. Physiological parameter changes, like HR and SpO_2_, are established assessment indicators of how well these manipulations were performed [15].

During the NICU stay, a neonate undergoes an average of 768 handling manipulations and 1341 invasive procedures, with manipulations accounting for 2 h and 26 min over 24 h [17]. Variation in physiological parameters during manipulations and procedures may be associated with negative health consequences. The fast-paced, stressful NICU environment may adversely impact how manipulations are performed, which may not be captured in procedure documentation in an EMR [18]. Recent studies have attempted to capture neonatal video streams by positioning a camera on the top of the neonate’s crib to overcome manual documentation limitations [19,20,21]. Along with the video streams, physiological data related to these manipulations can be simultaneously captured. Common non-invasive manipulations performed on neonates during NICU hospitalizations include patting, diaper change, and feeding. The decrease in SpO_2_ and bradycardia (less than 80 beats per minute) have been demonstrated before, during, and after diaper changes [21]. Neonatal comforting behaviors, such as patting, rubbing, holding, and stroking behavior, by nurses have also been studied using videotape analysis and were found to be irregular and associated with prolonged periods of neonatal distress [22]. Similar videotaped studies have found a lack of cue or infant-driven feeding methods used in neonates in the NICU [23,24].

Currently available public integrated ICU databases, like MIMIC, do not store video data of neonatal manipulation behaviors and synchronized physiological data. Appendix A, Table A1 outlines a detailed literature review of studies using video data along with physiological signal variations. There is a need to acquire continuous collated video data of manipulations and associated physiological parameters over the entire NICU stay to better assess how manipulations impact neonatal care outcomes. This data acquisition approach needs to address two essential design requirements. The first requirement is the millisecond resolution-based synchronization of captured video frames with physiological data captured from medical devices, which will enable an analysis of manipulation results related to medical events, such as apnea, desaturation, and bradycardia. The second key requirement is to automate data on salient features of manipulation into a patient’s EMR. Decreasing documentation demands using automatic monitoring and data-tagging may promote better nursing workflow and well-being. 

This study describes the acquisition and synchronization of video and physiological data in the NICU environment. We then present a machine learning (ML)-based automated tagging framework for three common neonatal manipulations: patting, diaper change, and tube feeding. Lastly, we demonstrate the value of synchronized video and physiological data by describing variations in physiological parameters associated with the identified manipulations.

## 2. Materials and Methods

This section describes the methodology of acquiring, synchronizing, and analyzing neonatal NICU data captured with respect to manipulations. 

### 2.1. Setting and Study Sample

Digital data were collected from a sample of neonates admitted to two NICUs over a three-month (April 2020–June 2020) duration. The study sites included 22 urban beds urban and 17 rural beds; both were level III NICUs in India. The urban NICU is staffed by three neonatologists with a doctorate in neonatal sciences, three residents, and 20 nurses. The rural NICU is staffed by three neonatologists with a doctorate in neonatal sciences, four residents, and 18 nurses. The Institutional Review Board of both NICUs approved the study with a waiver of informed consent. All electronic health records were de-identified (in accordance with Health Insurance Portability and Accountability Act (HIPAA)), and all the research was performed according to relevant guidelines. Prior to the study, written consent to the video monitoring and physiological data acquisition were obtained from the parents of eligible neonates at both study sites. All the data were stored in the de-identified form in the protected health information environment as per the HIPAA compliance. Hemodynamically stable neonates who stayed in the NICU for more than 24 h and did not have assisted ventilation were eligible. Neonates with congenital anomalies or on palliative care were excluded.

### 2.2. Data Collection 

A sample of 10 neonates was recruited for this study. De-identified individual patient admission-to-discharge data were electronically recorded using the iNICU platform [25]. This study was purely observational, and at no point in time were clinical decisions or interventions affected by study data results. The data were entered on bedside tablets through an iPad Pro (12.9 inches, IInd generation) using a Chrome browser, and data were stored in the Postgres SQL database. The clinical diagnoses of each neonate were determined by consulting neonatologists using the International Classification Diseases (ICD) ninth revision during daily rounds (morning, afternoon, and evening) performed at the patient bedside.

### 2.3. Video Acquisition of Manipulation 

During the study, the physiological data of neonates were collected using the NEO device [26]. The NEO system was improved with an additional camera module, and the size was further reduced (Appendix B, section B: NEO TINY system). Figure 1 shows the setup in a typical NICU setting. The wall mount was installed at the same height as the baby warmer’s top to minimize interference in the routine NICU workflow (Appendix B, Figure A1). The installed wall mount could be adjusted as per the discretion of onsite clinicians. The ‘Logitech C920′ Universal Serial Bus (USB) camera was installed facing the neonate. All the units’ beds were handled in the same way, and all the beds were equipped with cameras. The camera videos had a resolution of 1280 × 720 pixels and were recorded at 30 frames per second. 

Videos recording was continuous for most neonate’s NICU stay, but the parents or clinical staff could switch off the recording while the neonate was removed from the bed, such as during weight measurement and kangaroo care or breastfeeding, for privacy reasons. Thus, intermittent video data segments of each neonate were available for further analysis. 

### 2.4. Physiological Parameters of Manipulation 

Along with live video recording, real-time physiological data were simultaneously captured from the patient monitors (Appendix B: section D). All the monitors did not have the ability to record respiratory rate (RR); hence, this parameter was not used in the analysis. Heart rate (HR) and oxygen saturation (SpO_2_) were continuously recorded before, during, and after the manipulations.

### 2.5. Selection of Manipulations to Be Studied 

Video data were annotated manually with clinicians’ help, and a spreadsheet was maintained for ground truth labels of the manipulations. The overall system architecture is presented in the flow diagram shown in Figure 2. Appendix B describes the (A) hardware, data acquisition, and synchronization of video and physiological data and (B) software specifications. Appendix C describes the clinical staff interface to show an annotated video frame with physiological signals, missing data in the NICU environment, and data security. For the current feasibility study, we chose commonly used non-invasive manipulations (i) patting, (ii) diaper change, and (iii) tube feeding (definitions Table 1). The interventions were selected post hoc.

### 2.6. Input Data, Training, and Validation Data Set 

Examples of video captured patting, diaper change and tube feeding manipulations are shown in Figure 3. Acquired video sequences were down-sampled at 15 frames per second (fps) to reduce redundant computations, and images were resized from the original 1280 × 720 pixels to a color image of 720 × 480 pixels. Manipulations were initially divided based on category and neonatal identifier. Based on the discussions with the clinical team, it was hypothesized that 8 s of video data for any neonatal manipulation were sufficient to distinguish between the different types. Therefore, for each manipulation, data were processed at 8-s intervals amounting to 120 frames total. After that, the video clip corresponding to manipulation was extracted manually and then considered a training sequence. Following this, the next video sequence was extracted by sliding the cursor programmatically by 1 s to build the next 8 s subset. Although only the first 8 s were used for classifying the type of manipulation, all the frames were used for activity recognition. This process was repeated for the entire duration of the captured video of each manipulation. Appendix C explains how the clinical team visualized the video and physiological data.

### 2.7. Classification of Manipulation Using Convolutional Neural Network (CNN) 

The image classification technique has matured to a stage where facial recognition has become part of all consumer phones. An industrial set of algorithms trained on the large existing dataset is now available, which can be used to detect different images as per specific business domain requirements. In the current study (Figure 4), an existing pre-trained Inception-v3 CNN model [30] was used with prior ImageNet weights for colored Red Green Blue (RGB) images. The CNN-based models were then further improved with the concept of transfer learning [31], wherein the output of pre-trained models (such as InceptionV3) is trained for a specific task at hand. In our study, the task was to recognize the neonatal manipulations, and, currently, there are no established neonatal databases for neonatal procedures. We conducted the transfer learning process by providing training on our annotated images marked as (i) patting, (ii) diaper change, and (iii) tube feeding. This step improves the accuracy of the manipulation-tagging model. 

The performance of the InceptionV3 CNN model with the transfer learning layer was also visualized by the t-Distributed Stochastic Neighbor Embedding (t-SNE) plot [31,32], which take perplexity as a user-specified input parameter. Perplexity corresponds to the effective number of neighbors considered for obtaining the embeddings and was shown to be robust over the range of 5–50 [33]. We picked the perplexity value of 35 to visualize the best segregation of neonatal manipulation. The individual image frames of videos were resized to 226 × 226 pixels as per Inception-v3 specifications.

### 2.8. Activity Recognition Combining CNN Output with LSTM

From a computer vision perspective, a neonatal manipulation, such as diaper change, is a collection of image frames collected over time encapsulating the activity (manipulations). Therefore, we further wrap up the pre-trained CNN model into a time series layer to bring the concept of manipulation (sequence of images). The output of the Time-distributed CNN model generates an output of the 2048-dimensional feature vector. This vector conveys information about constituent objects, such as the neonate, the clinical staff, diapers, syringe, and plunger, and their spatial attributes and how they correlate during the manipulations. It is not feasible to visualize these vectors in a human-readable format in the current deep learning landscape.

The CNN models are very accurate in classifying images, but other branches of machine learning, such as deep learning (e.g., Long Short Term Memory; LSTM), have also progressed to identify the activities. After training of the combined CNN and LSTM, the system can automatically classify the neonatal manipulations.

We extracted the weights of the CNN (InceptionV3) model to extract features of the images and combine them with LSTM layers to perform activity recognition. The sequential 2048 feature vector, an output of the InceptionV3 model representing activity in a manipulation, was input to the LSTM model. The LSTM layers were followed by additional dense layers and followed by a three-layer softmax layer. An early stopping criterion with the patience of 8 was employed. This monitors the validation loss and stops the training when the loss deteriorates for eight successive epochs. The model was implemented in Keras [34] and TensorFlow [35] and used the ‘categorical cross-entropy’ loss function and ‘adam’ optimizer. The EarlyStopping callback was used to stop training on the epoch when the accuracy metric has stopped improving [36].

### 2.9. Variation in Physiological Signals Associated with Manipulation

The variations in physiological parameters during manipulations were compared with those of baseline (defined as 5 min before the manipulation) and post-manipulation (defined as 5 min after the manipulation). 

### 2.10. Performance Metrics 

We measured the performance of the CNN/LSTM model in the classification of neonatal manipulations using Positive Predictive Value (PPV) (Equation (1)), Sensitivity (Equation (2)), and F-measure (Equation (3)), which are defined as:(1)$$\mathrm{PPV}=\frac{\mathrm{TP}}{\mathrm{TP}+\mathrm{FP}}$$
(2)$$\mathrm{Sensitivity}=\frac{\mathrm{TP}}{\mathrm{TP}+\mathrm{FN}}$$
(3)$$F-\mathrm{measure}\;=\frac{2\times \left(\mathrm{PPV}\times \mathrm{Sensitivity}\right)}{\mathrm{PPV}+\mathrm{Sensitivity}}$$
where TP, FP, and FN are true positive (TP: manipulation patting, diaper change, and tube feeding detected correctly), false positive (FP: when the system detects a manipulation when there is none), and false negative (FN: when there is manipulation that the system does not detect). For data with normal distribution, a two-sided paired t-test with a significance level <0.05 was used to compare physiological parameters during and after manipulations. This was based on our assumption that the physiological values may increase or decrease during and after manipulations in comparison to the baseline data.

### 2.11. Overall Activity Detection Model Evaluation

The model evaluation was done using leave-one-out cross-validation (LOOCV) utilizing PPV and sensitivity metrics. In the NICUs involved in the current study, nurses did not document routine care activities, such as diaper change and patting, in the EMR system. The comparison of tube feeding records between documented nursing records and automated tube feeding notes highlights the additional temporal data captured by machine learning-based automated classification system. The tube feeding duration and time duration from the last tube feeding were not captured in current EMR records. 

Based on the visual investigation of data with the clinical team, spatial and temporal features in manipulations were documented (Table 1) to understand the classification task.

## 3. Results

The results of the feasibility study conducted to verify the designs of automated tagging of manipulation are below.

### 3.1. Baseline Data

Ten neonates admitted to NICU were enrolled from December 2019 to April 2020. The baseline characteristics of study subjects are displayed in Table 2. The mean gestational age was 34.7 weeks (range, 26 weeks to 40 weeks), and the mean birth weight of study subjects was 1893.8 g (range, 800 g to 3231 g). 

### 3.2. Distribution of Manipulations

Table 3 shows the average duration of a patting, diaper change, and tube feeding. A total of 64 diaper changes (average duration, 45.5 s), 108 tube feedings (average duration, 108.9 s), and 167 patting’s (average duration, 28.9 s) were recorded and utilized for analysis. 

### 3.3. CNN Based Classification of Manipulations

The 2048 features generated from manipulation images were plotted using t-SNE visualization (a) without transfer learning, which means without the knowledge of the current domain, and (b) with transfer learning. Without the transfer learning (Figure 5a), the ImageNet based Inception-V3 pre-trained model cannot classify the neonatal manipulations. However, after the transfer learning, except for a few outliers, the transfer-learning based Inception-V3 model can visualize the images of patting, diaper change, and tube feeding successfully (Figure 5b). 

The accuracy of CNN-based model in classifying the manipulation frame/image is displayed in Figure 6. The validation accuracy (red) was achieved after eight epochs.

### 3.4. LSTM Based Classification of Manipulation Videos

The validation of automatic video classification was done in clinical settings, and the accuracy was 85% on the validation dataset. The comparison of NTS data with respect to nurse documented procedures is shown in Table 4. The 2048 features from the Inception-V3 model were generated for all frames present in the duration of the manipulation video. 

The performance of the deep learning model obtained is presented in Table 5. The model automatically annotates the manipulation of a given neonate. Figure 7 demonstrates different manipulations that are classified by the CNN/ LSTM model during the validation phase.

### 3.5. Physiological Signal Variations during Manipulations

Figure 8a–c show variations in physiological parameters during the patting, diaper change, and tube feeding manipulations, respectively. There was an associated increase in normalized heart rate between before and during the period for neonates <32 weeks’ gestation (shown blue color) for all the three manipulations.

Table 6 shows the HR and SpO_2_ physiological variables for each of the three manipulations. The significant changes (*p* < 0.05) are: (I)For <32 weeks: (a) HR increased during diaper changes and decreased afterward, (b) SpO_2_ increased during the diaper change.(II)For ≥32 weeks: (a) HR increased during patting and decreased afterward, (b) the HR decreased after tube feeding.

## 4. Discussion

The NICU environment is highly complex, with critically ill neonates who require multiple medical devices, such as patient monitors, ventilators, syringe pumps, and infusion pumps. These many devices leave minimal working space for movement around the bedside. Therefore, a pocket-sized data aggregator, NTS, has been developed to capture valuable data with a small footprint; with its pocket-sized design (5.8 cm × 4.1 cm × 7.7 cm), it is ideal for cluttered workspaces and roaming device workflows. For video monitoring, the camera was wall-mounted above the neonate’s bed to avoid interfering with routine workflow in the NICU. The NTS client device synchronizes the acquired medical device and video data and sends it to the EMR platform. The platform displays the live video feed of a neonate, along with all the acquired vital parameters data for clinical interpretation. 

The framework presented in this study can enable automatic identification of manipulation, generate corresponding EMR documentation of those manipulations, and measure changes in physiological parameters. The study demonstrates a machine learning model to classify three common neonatal care manipulations: (a) patting, (b) diaper change, and (c) tube feeding. It is important to highlight that the transfer learning of classifying the manipulations like tube feeding will strongly depend on local practices, such as syringe use, the position of the end for the tube, and even the use of gloves (and their colors). The authors anticipate that NICUs in a given geographical region or associated with similar neonatal research networks can develop a unique dataset of images as per their practices. This dataset can be readily used as ‘training’ module for the system for that group of NICU units. 

In this study, the model was able to classify the manipulations with 95% accuracy in the training dataset (accompanying loss of 0.0026) and 85% in the validation dataset (with accompanying loss of 0.0409). During the manipulations, the physiological parameters were compared with those captured prior to the manipulation and after the manipulation, in neonates <32 weeks’ gestation, diaper changes were associated with significant changes in HR and SpO_2_ (perhaps due to crying with subsequently increased minute ventilation). In comparison, for neonates ≥32 weeks’ gestation, patting and tube feeding was associated with significant changes in HR. The health impact of these vital sign changes associated with routine care practices is unclear. The ability to detect continuous changes in physiological parameters associated with machine learning-driven monitoring of common neonatal manipulations in the NICU illustrates the capability of the NTS model, which could be further used for further analysis of how neonatal manipulations and procedures impact short- and long-term outcomes. 

Most NICUs have strict light and sound control protocols, both in the larger NICU environment and in the local environment of each neonate. In the current study, open incubators were used with most of the neonates. The lights in the NICU were recommended to be dim most of the time. We did not find any difference in the automatic classification of manipulations in different light conditions. However, these finding needs to be confirmed with a large sample size. In future studies, the feasibility of night vision mode in these cameras needs to be explored in poor light conditions. Moreover, the advent of 3-D cameras allows manipulation of specific data to be captured, which will also be explored in future efforts. With the emergence of artificial intelligence, it is anticipated that continuous monitoring and analysis will help avoid unnecessary manipulations that may cause a negative neuro-sensorial stimulus to premature and sick neonates. If specific neonatal manipulations and procedures are associated with worse outcomes, future research using the NTS model could assess how modifying routine care practices to target vital sign ranges could improve outcomes.

## 5. Limitations

While the presented study shows promise for future NICU neonatal monitoring applications, certain limitations need to be considered. As a pilot study to assess the feasibility of the system, only a small number of patients were recruited. Future studies will need to assess potential differences regarding gender, different gestational age groups, and other demographic parameters. A larger cohort of neonates will need to be recruited to build a physiological database that will provide more balanced data for machine learning models to simulate the NICU environment. The presented approach only utilized labeled data of three manipulations for ten neonates. The recognition capabilities of the deep learning model can be explored further by including the data of more manipulations and more neonates. (e.g., some neonatal manipulations or procedures, such as heel prick, last only a few seconds). In the current study, monitors did not capture per second data; hence the study lacks the complete resolution of physiological data required for the detailed analysis of brief manipulations or procedures. The study did not consider the medications that neonates were receiving during their stay in the NICU; since sedatives and analgesics can potentially affect the stress experienced by neonates [37] future studies should consider individual patient drug dosages and half-lives.

## 6. Conclusions and Future Directions

The present study demonstrates a framework to help clinical staff evaluate changes in physiological parameters associated with common care manipulations in the NICU. Due to the limitations of human resources, close and constant observation of neonates on a 24-h basis is a challenge. The current study model, which utilizes state-of-the-art computer vision and analyzes physiological parameter variations, may be a useful adjunct to assess neonates. Moreover, this framework will be extended to build video databases for other neonatal manipulations and procedures, which can be used for (a) skill evaluation of clinical staff and (b) improving the care documentation. Although the current results showed the feasibility of the system, its efficiency still needs to be studied in the larger NICU population across different sites. Another future direction is to include surrounding contextual data, such as lighting conditions, ambient noise in the NICU, and the number of clinical staff around neonates, to study the overall effect on the neonates while conducting manipulations.

Future studies will capture real-time physiological data from bedside monitors in millisecond resolution synchronized with the video data. The millisecond data will help study the impact of non-invasive and invasive manipulations (such as heel prick, intubation, and extubation) in a more granular manner with associated clinical events apnea, bradycardia, and desaturations. Recent advances in the computer vision and deep learning community have shown successful use of semi-supervised and unsupervised domain adaptation techniques. These methods could be leveraged to reduce the data labeling requirements further, while adapting the proposed system to new NICU units. In addition, given reported racial disparities in neonatal care in the United States [38,39], the NTS system could be used to study racial inequities in the NICU regarding average time dedicated to care manipulations of neonates from different racial backgrounds to provide quantifiable, informative data to healthcare providers.

## 7. Code Availability 

The code that underpins the video analytics documentation is openly available. A Jupyter Notebook containing the code used to generate the descriptive statistics and tables included in this paper is available at: https://github.com/CHIResearch/IEEEVideo. README.md file has all the script-related and other details. 

## Figures and Tables

**Figure 1 children-08-00001-f001:**
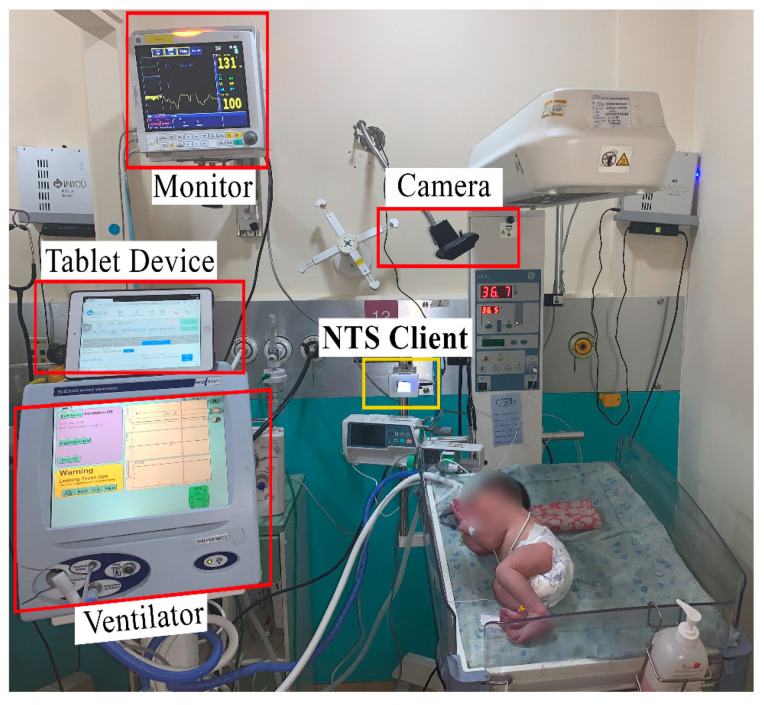
NEO TINY system (NTS) client module in typical Neonatal Intensive Care Unit (NICU) settings (box with yellow-colored border highlight the NTS client, and red-colored boxes highlight other devices).

**Figure 2 children-08-00001-f002:**
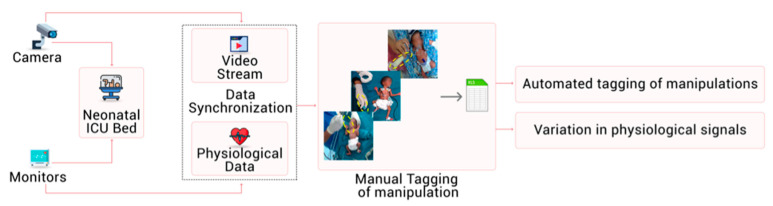
The overall architecture of machine learning (ML)-based video classification system in the NICU.

**Figure 3 children-08-00001-f003:**
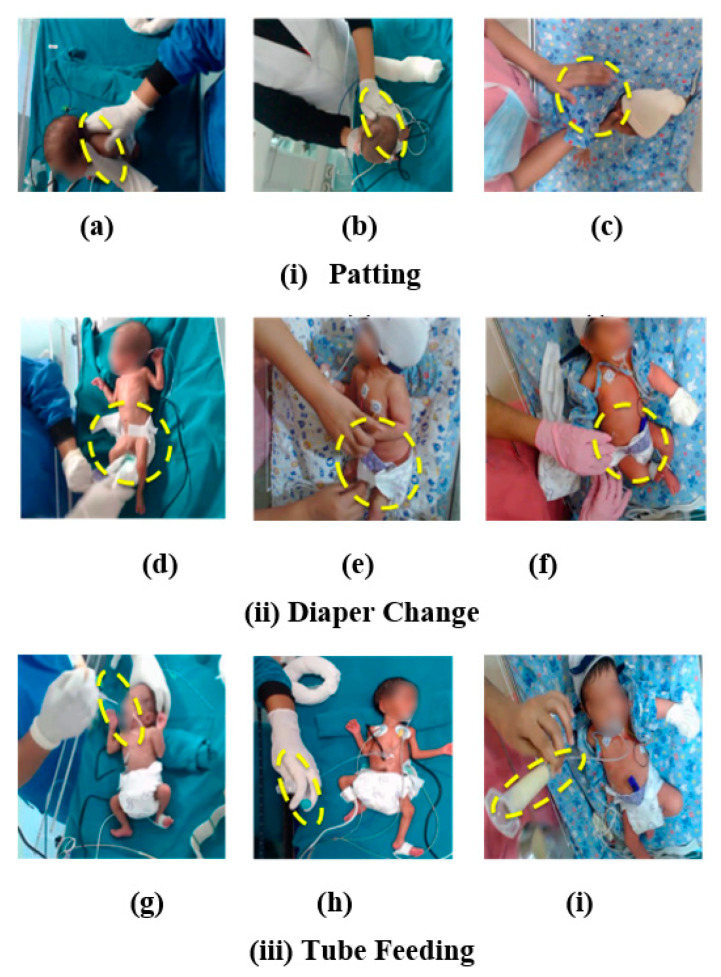
Images of manipulation: (**i**) patting, (**ii**) diaper change, and (**iii**) tube feeding, the region of interest marked with a yellow border.

**Figure 4 children-08-00001-f004:**
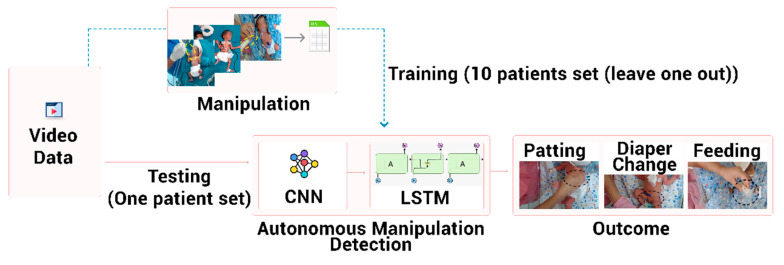
Deep learning architecture for neonatal video classification utilizing Convolutional Neural Network (CNN) and Long Short Term Memory (LSTM) network.

**Figure 5 children-08-00001-f005:**
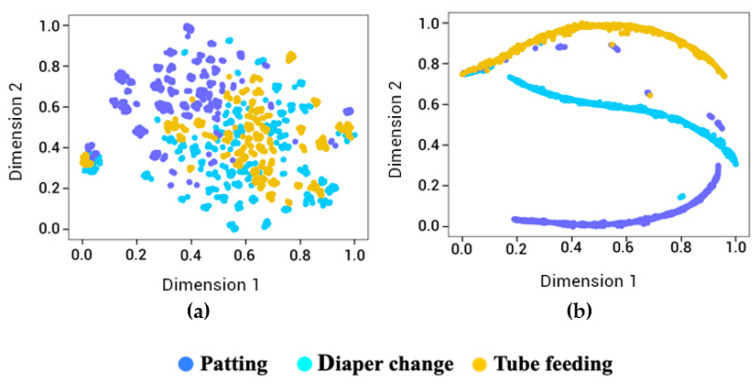
t-Distributed Stochastic Neighbor Embedding (t-SNE) visualization for the manipulations (patting, diaper change, and tube feeding) (**a**) without transfer learning and (**b**) with transfer learning. Perplexity is 35, and the number of iterations is 20,000.

**Figure 6 children-08-00001-f006:**
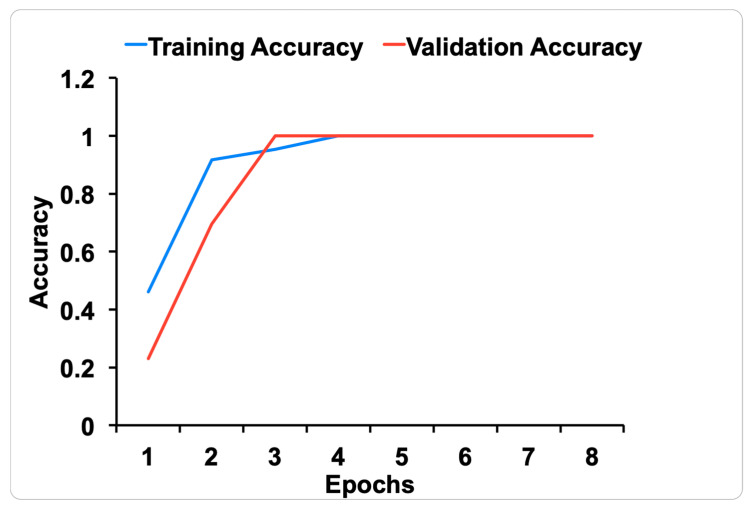
CNN-based model accuracy for classifying manipulation images.

**Figure 7 children-08-00001-f007:**
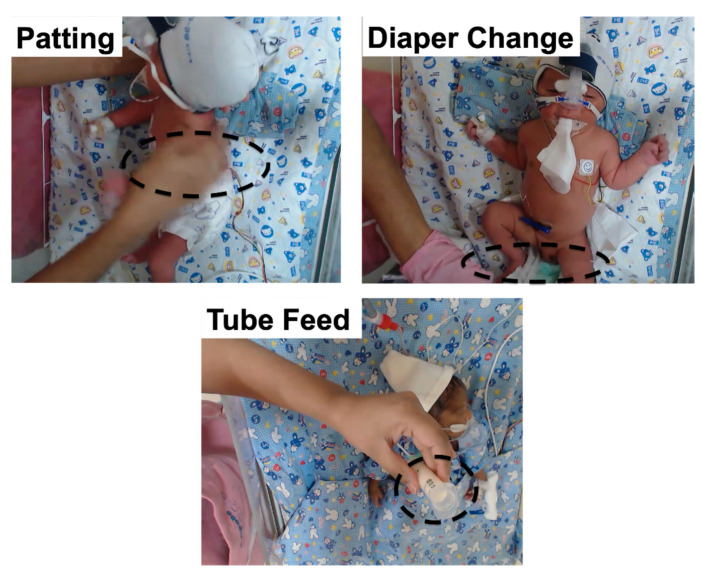
Automatic tagging of manipulation videos: The first frame identified as manipulation is marked on the top left, and dotted lines show manipulation.

**Figure 8 children-08-00001-f008:**
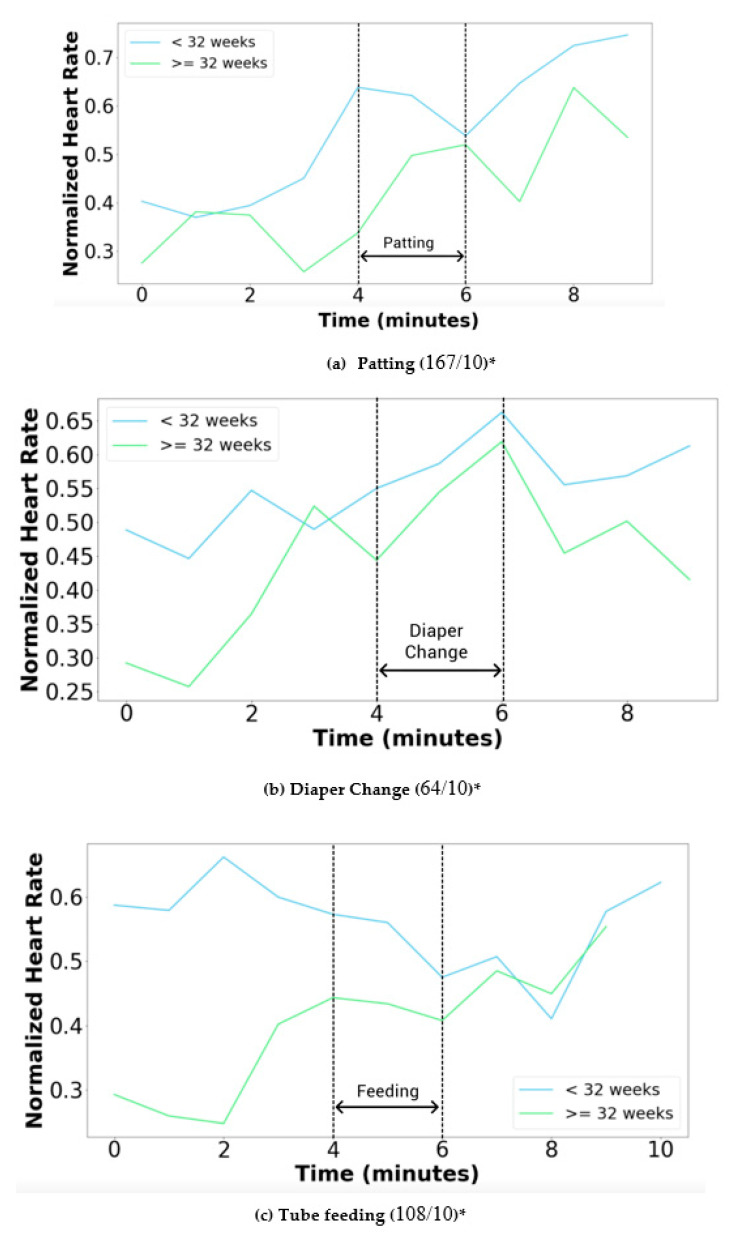
Variability in physiological signals captured every minute (average values) during the manipulations in the clinical setting. (**a**) patting, (**b**) diaper change, and (**c**) tube feeding. * number of manipulations/number of patients.

**Table 1 children-08-00001-t001:** Visual features of manipulations.

Manipulation:	Characteristics	Ref
Patting:	Definition: This is a comforting manipulation where the flat surface of the palmer surface of the caregiver’s hand was brought into contact with a surface of the neonate’s body singly or repetitively. The intensity and rate were variable in different episodes of patting.	[22]
	Spatial features: Nurse’s hand, neonate’s body boundaries	
	Temporal features: Frequency: On-demand Duration: 33 s	
Diaper Change:	Definition: This manipulation involves changing the diaper and cleaning the diaper area for skin hygiene.	[27,28]
	Spatial features: Two nurse’s hands, diaper, and skin contrast	
	Temporal features: Frequency: 4 h Duration: 3 min	
Tube Feeding:	Definition: This manipulation utilizes a soft tube placed through the nose (nasogastric) or mouth (orogastric) placed into the stomach. The feeding is provided through a tube into the stomach until the baby can take food by mouth.	[29]
	Spatial features: Nurse’s hand, milk, syringe attached to the feeding tube (with or without plunger)	
	Temporal features: Frequency: 2 h Duration: 10–30 min	

**Table 2 children-08-00001-t002:** Baseline characteristics of the sample (enrolled subjects, *n* = 10).

Id	Sex	Gestational Age	Birth Weight (g)	Age Interval for Recording (Days)	Clinical Diagnoses
1	Male	26^+0^	1005	24–25	RDS, Apnea, Prematurity
2	Male	27^+1^	800	76–90	Prematurity
3	Male	29^+4^	1372	37–44	Prematurity, RDS, Apnea Sepsis
4	Male	35^+2^	1400	8–10	NNH
5	Male	36^+0^	2400	3–5	RDS, NNH
6	Male	36^+6^	1430	4–8	Prematurity, NNH
7	Male	36^+6^	3231	5–6	RDS
8	Male	39^+2^	2600	7–8	RDS, Seizure
9	Male	39^+4^	2000	5–6	Sepsis, RDS, Apnea
10	Male	40^+0^	2700	3–7	RDS, NNH

RDS: Respiratory Distress Syndrome, NNH: Neonatal Hyperbilirubinemia.

**Table 3 children-08-00001-t003:** Frequency and duration of manipulations recorded.

Manipulation	^#^ Frequency	* Average Duration (Seconds)	Minimum Duration (Seconds)	Maximum Duration (Seconds)
Patting	167	28.9 (12.4)	12	56
Tube Feeding	108	108.9 (55.3)	25	300
Diaper Change	64	45.5 (18.8)	17	92

^#^ Frequency across the length of stay in NICU, * Mean (Standard Deviation).

**Table 4 children-08-00001-t004:** NTS generated note of neonatal manipulations.

**Patting**	Nurse	Not Captured in EMR
	NTS	The patting was started at 14:05:08 on 17-08-2020 and completed at 14:06:19 (duration: 71 s). This is manipulation number 3, since 8 a.m.
**Diaper Change**	Nurse	Not captured in EMR
	NTS	The diaper change was started at 19:35:25 on 17-08-2020 and completed at 19:37:01 (duration: 96 s). This is manipulation number 4 since 8 a.m.
**Tube feed Entry**	Nurse	Start Time: 17-08-2020 09:30 a.m. Type: Tube Feed Type of Milk: Preterm Formula Quantity: 11 mL
	NTS	The feeding was started at 09:30:09 on 17-08-2020 and completed at 09:32:57 (duration: 168 s). This is manipulation number 1 since 8 a.m.

**Table 5 children-08-00001-t005:** Performance of deep learning model.

	PPV	Sensitivity	F-Measure	Total Manipulations
Patting	0.86	1.00	0.92	167
Diaper Change	0.98	0.68	0.80	64
Tube feeding	1.00	0.87	0.93	108

**Table 6 children-08-00001-t006:** Physiological parameters (HR and SpO_2_) before, during, and after manipulation.

		<32 Weeks					≥32 Weeks				
Manipulations	Parameters	Baseline *	During *	Post *	*p*-Value ^$^	*p*-Value ^#^	Baseline *	During *	Post *	*p*-Value ^$^	*p*-Value ^#^
Patting	HR (BPM)	161.9 (10.19)	164.7 (13.7)	157.6 (24.9)	0.168	0.069	148.7 (13.9)	165.7 (30.7)	150.9 (8.2)	0.019	0.00
	SpO_2_ (%)	92.7 (7.4)	93.0 (7.9)	89.7 (12.8)	0.43	0.087	94.7 (6.1)	92.5 (10.9)	93.5 (11.41)	0.21	0.34
Diaper Change	HR (BPM)	152.738 (31.4)	166.9 (14.4)	157.4 (23.2)	0.000	0.036	147.8 (12.02)	152.7 (15.8)	150.7 (9.5)	0.10	0.17
	SpO_2_ (%)	88.9 (18.2)	94.02 (5.7)	89.4 (13.6)	0.000	0.07	94.7 (5.8)	94.9 (5.4)	93.9 (12.7)	0.44	0.36
Tube Feeding	HR (BPM)	163.1 (10.55)	164.28 (13.29)	162.2 (20.0)	0.26	0.22	150.5 (16.7)	147.6 (16.6)	153.3 (11.6)	0.17	0.003
	SpO_2_ (%)	93.9 (6.4)	93.9 (4.9)	91.7 (9.9)	0.49	0.052	95.1 (4.9)	94.0 (8.0)	93.5 (7.6)	0.23	0.37

* Mean (Standard Deviation); HR: Heart rate, SpO_2_: Oxygen saturation, BPM: Beats per minute; ^$^ Comparing baseline and during manipulation parameters, ^#^ Comparing parameters during manipulation and post manipulation.

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
