# Peer review of "Machine Learning-Based Automatic Classification of Video Recorded Neonatal Manipulations and Associated Physiological Parameters: A Feasibility Study"

_children, 2020, doi:10.3390/children8010001_

Round 1
Reviewer 1 Report
Thank you for the opportunity to review the manuscript children-1024194 "Physiological parameters associated with autonomous monitoring of common neonatal manipulations".
The manuscript describes the feasibility assessment of a novel, up-to-date, high technology approach to detect, record and analyse neonatal procedures synchronously with the usual parameters of physiologic monitoring. Digital technology, data science and machine learning are used to develop a proposal for complementary autonomous monitoring in the NICU setting. It is, in fact, a research with potential high interest for Paediatrics in general and Neonatology in particular.
Presenting high technology information to a clinical audience using a clinical journal is a hard challenge. The Authors are advised to consider the revision of the manuscript (and the study) to improve it, as a form to reach more scientific soundness in the chosen field of dissemination.
The presented title induces the reader to find something quite different from the real content. The reader has to wait until the Conclusion to find what the study is really about: “the current results showed the feasibility of the system” (line 348). The manuscript will benefit it the title already informed that a feasibility study is presented. Even so, the present title would be miss phrased, as the target "physiological parameters” are indeed “associated with “common neonatal manipulations", not with “autonomous monitoring ”.
The effort of the tech experts in the author’s team is acknowledgeable and the choice to use the appendices for deeper characterization of the technology is wise. An even greater effort must nevertheless be taken, as some “technicalities” could be made easier for the paediatric reader to understand. The most important issue is the terminology used to refer to the parameters used for assessing the performance of the proposed “autonomous monitoring ”. The Authors must present as methodology all the performance parameters used, defining them as needed, and either applying the terminology commonly used in clinical epidemiology or an equivalence of terms must be presented; some examples are pointed out in the annotations of the manuscript.
There are consensual norms to refer the devices and the software used in the studies, which development is not the main issue of the manuscript. The Authors are advised to follow those norms.
The study assessed the novel technology in the setting of a single, unidentified NICU, applying the technology on recruited subjects. Those recruited neonates must be referred to as a sample; the criteria for eligibility must be presented, as are the inclusion as exclusion criteria. Heterogeneous as the neonatal population is, the sample must be correctly described in its demographic and clinic characteristics; as it presently is, the description is insufficient; some examples are pointed out in the annotations of the manuscript.
The description of both the manipulations and the physiological parameters are better described independently from the demographic and clinic characteristics of the recruited sample. The differential presentation and analysis of data from neonates born at les or more than 32 weeks of gestation is nevertheless correct, undisputed and necessary.
The Authors are strongly advised to review all the statists of the manuscript, both as presented in Methods and as described to have been applied in Results. Some examples of inadequate statistical description of variables (both by numerical parameters and graphically) are pointed out in the annotations of the manuscript. The statistic tests used for data analysis are very insufficiently described in the Methodology section and, unfortunately, the inadequate statistical description of variables easily induces to believe that the inadequate testing was also applied. These grounded doubts stain the study mercilessly. Those statistic errors are gladly easy to correct with the adequate epidemiologic and statistic assistance.
For specific reviews and recommendations, please refer to the annotations of the manuscript.
I recommend considering in the Discussion a major characteristic of most NICU: the effort to spare immature and sick neonates unnecessary and potentially harmful neurossensorial stimuli. Most NICU have strict light and sound control, both in the larger NICU environment and the closer, own environment of each neonate. Open incubators, as shown in the manuscript, are far from being the most used; closed incubators are widely used and they are very frequently under almost opaque covers; moreover, lights in the NICU are advised to be dim most of the time. How do these conditions affect the feasibility and the effective performance of the proposed “autonomous monitoring” requires discussion from the Authors.
Finally, the authors come from a wide variety of institutions and fields, from the technology industry and services to Academia and neonatal care. The quality of the interests of each author and their institutions on the research are certainly different, as some of them present data about the performance of commercial products they developed (and certainly wish to commercialize it), and others provided a clinical setting to assess the devices. One would like to view a clear presentation of the apparent conflicts of interests that may be present. Financial issues among the authors and their institutions, if present, should be acknowledged; independent data analysis must be asserted.

Reviewer 2 Report
What do the authors mean with autonomous monitoring: it is in the title and the abstract, the intention is ‘tagging’ of neonatal manipulations = please ‘sell’ what you wanted to do, and have done better. It is only when reading the full paper that this becomes clearer. (I initially anticipated to read on autonomic nervous system aspects during care).
Why have these interventions been ‘selected’, was the predefined based on a study protocol, or a post hoc, during analysis decision ?
I anticipated issues like eg kangaroo care, of skin breaking procedures, while patting reads bizar ?
It is not clear to this reviewer how the consent and study registration has been conducted ? what do the authors mean with ‘authorizied’ body, informed consent (signed ? blinded data, how were data handled and stored). In my assessment, I do have concerns on the ethics as described at present (not the absence of an intervention, but data acquisition).
Were all beds of the units handled in the same way, as this likely will also affect case mix, and videos were recorded at random intervals: how should I assess this, and is this also a potential bias (recordings over the full length of stay, or has ‘sampling’ be distorded ?)
If you look at figure 3, it is my opinion (clinical neonatologist, have been working in different countries and units), that the ‘learning’ on eg tube feeding will strongly depend on local practices like eg syringe use, position of the (end) for the tube, and even the use of gloves (and their colours). So yes, likely feasible, but extrapolation is a major issue, or do you intend to develop an ‘training module for the system’ for each unit ? I’m quite confident that a validation effort in another unit likely will result in even poorer results.
What’s the rationale of a frame of 8 seconds ? have other frames been considered ?
Where the changes in oxygenation/saturation ‘true’ or in themselves, signal capture related (so has the quality of the signal been assessed)
Minor
The quality of the pictures should be further improved.
In essence, this paper suggests that in a pilot system, specific procedures in a specific unit can be recognized by machine learning concept.
Round 2
Reviewer 1 Report
Thank you for the opportunity to review the new, revised version of the manuscript children-1024194 "Physiological parameters associated with autonomous monitoring of common neonatal manipulations", with its new title "Machine learning based automatic classification of video recorded neonatal manipulations and associated physiological parameters: A Feasibility Study".
I have to acknowledge the effort made by the authors complying with the reviewers suggestions and very much improving the manuscript. I am now very pleased with the manuscript.
I would yet like to see an homogeneous correction of grammar related to the plural noun "data" (singular "datum"). In some sentences it is correctly written "data were" but in many sentences there is still incorrectly written "data was".
Reviewer 2 Report
no additional comments
Author Response
Authors thank reviewers and editor in chief for considering our paper and sharing valuable comments. Below is comment-by-comment response:
Answer to reviewer’s comments:
no additional comments
Response: Thank you for considering the manuscript.